# Using Social Network Analysis to Identify Spatiotemporal Spread Patterns of COVID-19 around the World: Online Dashboard Development

**DOI:** 10.3390/ijerph18052461

**Published:** 2021-03-03

**Authors:** Kyent-Yon Yie, Tsair-Wei Chien, Yu-Tsen Yeh, Willy Chou, Shih-Bin Su

**Affiliations:** 1Department of Gastrointestinal Hepatobiliary, Chi Mei Jiali Hospital, Tainan 700, Taiwan; kyawsein@gnail.com; 2Department of Medical Research, Chi-Mei Hospital, Tainan 700, Taiwan; smile@mail.chimei.org.tw; 3Medical School, St. George’s University of London, London SW17 0RE, UK; jess97yeh@gmail.com; 4Department of Physical Medicine and Rehabilitation, Chi Mei Medical Center, Tainan 700, Taiwan; 5Department of Occupational Medicine, Chi Mei Medical Center, Tainan 700, Taiwan

**Keywords:** COVID-19, social network analysis, item response model, correlation coefficient, daily confirmed case, spatiotemporal spread pattern

## Abstract

The COVID-19 pandemic has spread widely around the world. Many mathematical models have been proposed to investigate the inflection point (IP) and the spread pattern of COVID-19. However, no researchers have applied social network analysis (SNA) to cluster their characteristics. We aimed to illustrate the use of SNA to identify the spread clusters of COVID-19. Cumulative numbers of infected cases (CNICs) in countries/regions were downloaded from GitHub. The CNIC patterns were extracted from SNA based on CNICs between countries/regions. The item response model (IRT) was applied to create a general predictive model for each country/region. The IP days were obtained from the IRT model. The location parameters in continents, China, and the United States were compared. The results showed that (1) three clusters (255, n = 51, 130, and 74 in patterns from Eastern Asia and Europe to America) were separated using SNA, (2) China had a shorter mean IP and smaller mean location parameter than other counterparts, and (3) an online dashboard was used to display the clusters along with IP days for each country/region. Spatiotemporal spread patterns can be clustered using SNA and correlation coefficients (CCs). A dashboard with spread clusters and IP days is recommended to epidemiologists and researchers and is not limited to the COVID-19 pandemic.

## 1. Introduction

The COVID-19 pandemic is spreading widely around the world, causing significant threats to the public. As of February 20, the world has accumulated more than 0.11 billion the COVID-19 cases, threatening people’s health, economic development, and social stability [1]. Many online real- or near-real-time dashboards have been launched to track the worldwide spread of the COVID-19 outbreak [2] and to provide the cumulative numbers of infected cases (CNIC) daily to the public, such as the Johns Hopkins University Center for Systems Science and Engineering dashboard (JHU) [3], the Leszkiewicz personal dashboard [4], the World Health Organization dashboard [5], HealthMap [6], and the dashboard created by Schiffmann, an 18-year-old high school senior from Washington State in the United States [7].

Although those website owners [3,4,5,6,7] made efforts to launch dashboards, the sites merely display the basic information on the CNIC of COVID-19 with bubbles on a world map. Other websites [8,9,10,11,12] also provide common and regular information (e.g., with a traditional world map and CNIC or daily confirmed cases (DCC)) to the public. None of these websites are equipped with spatiotemporal spread patterns (SSP) of COVID-19 and the inflection points (IP) [13,14,15,16] to present the negative impact of COVID-19 (NISHC) on a world map using an overall score to fulfill the public’s interest. Displaying CNIC bubbles on world maps in areas is common during the COVID-19 pandemic. Determining how to display both features (i.e., SSP and IP days) on a world map is challenging. We were thus motivated to design a dashboard that can simultaneously display both SSP and IP days and complementarily characterize spread patterns of COVID-19 across the globe.

Furthermore, few authors are willing to provide their detailed experimental data to readers who are going to replicate the study on their own. Data, along with MP4 videos and modules in articles, would be an interesting resource and have rarely been seen before in the literature.

### 1.1. Social Network Analysis for Identifying Spread Patterns Based on DCC

Social network analysis (SNA) [17] is the process of investigating social structures through the use of networks and graph theory. It characterizes networked structures in terms of nodes (individuals, groups of people, or objects within the network) and the ties, edges, or links (relationships or interactions) that connect them. Examples of social structures commonly visualized through SNA can be found in many academic disciplines and fields [18,19,20,21,22,23]. The use of SNA was thus expected to identify SSP of the COVID-19 cases around the world.

### 1.2. Item Response Theory and the Infection Point on an Ogive Curve

Many methods [24,25,26,27,28,29,30,31,32] have been used to build mathematical models for predicting the CNIC or DCC of COVID-19. None of the previous studies applied the item response theory (IRT) [33,34] to the pandemic outbreak. The IRT model, using probability to display the ogive curve (based on CNIC), is emulated as the epidemic trend on the expected CNCI. Two parameters (i.e., location on the *x*-axis and slope as the steep of a curve toward the *y*-axis represented by item difficulty (b) and discrimination (a), respectively) are involved in a predictive model. The infected days on the *x*-axis are transformed to be the so-called ability parameter (θ) on a continuum scale from the left to the right side (e.g., in a range from −5 to 5) [32]. As such, the cumulative probabilities on the *y*-axis can be converted back to the expected CNIC for a specific country/region that has been impacted by COVID-19. The inflection point (IP) [13,14,15,16,32] can be determined on the IRT ogive curve accordingly.

### 1.3. Two Phenomena Observed in the COVID-19 Pandemic

We observed that the SARS-CoV-2 (COVID-19) disease began in Wuhan, China [35] and was spread to West Asia, Europe, North America, and South America. Whether the spread routes of COVID-19 are evident using SNA is worth studying.

After entering the winter season, starting in January, the deadly coronavirus experienced a new wave (called the second wave) of sporadic cluster cases, especially in colder cities. For example, the northern and northeastern cities in China, including Shijiazhuang City in Hebei Province and Suihua City in Heilongjiang Province, began to report new cluster cases [36]. Shijiazhuang City in Hebei Province has been listed as a high-risk region. Currently, Shijiazhuang City is the only high-risk region in China, with over 300 new confirmed cases, a record high in over 5 months, in the 10 days from 2 to 11 January 2021 [36]. Two questions were raised: (1) Whether both the provinces of Hebei and Heilongjiang in China have a similar spread pattern, and (2) which countries (or regions) have the same spread pattern as Hebei Province (China).

### 1.4. The Aims of This Study

Two parts are involved in this study: Identifying the spread patterns in the two periods (1) from 22 January to 27 March 2020, and (2) from 1 January to 16 February 2021. The aims of this study are to (1) illustrate the use of SNA for investigating the association of DCCs in regions; (2) examine the spread routes of COVID-19 from China to West Asia, Europe, North America, and South America; (3) identify countries/regions with the same spread pattern as Hebei Province in China; and (4) design a dashboard for better interpretation of CNIC and the corresponding IP days by country/region.

## 2. Materials and Methods

### 2.1. Data Source

We downloaded COVID-19 outbreak DCCs in the two periods (1) from 22 January to 27 March 2020, and (2) from 1 January to 16 February 2021, as mentioned above from GitHub [8], a site that provides information on newly DCCs in countries/regions around the world. All the downloaded data with 299 countries/regions are publicly displayed on the website. Ethical approval is not necessary for this study because all the data are obtained via the Internet (see Appendix A).

### 2.2. Spread Routes of COVID-19 across Continents

Although we knew that the spread routes of COVID-19 were from Asia to Europe and then North and South America, the spread routes are still unknown. In March 2020, the US Centers for Disease Control and Prevention (CDC) established geographic risk stratification criteria for the purpose of issuing travel health notices for countries with COVID-19 risk and guiding management decisions for people with potential travel exposures to COVID-19 [37]. The restriction on entry to the United States for foreign nationals from China and Iran was issued on 16 March 2020 or earlier. The lag-behind spread pattern is required for investigations on a different DCC pattern for COVID-19 between China and the US. The spread routes are thus tracked with SSP using SNA.

### 2.3. The Three Steps Below Were to Identify the Spread Patterns of COVID-19


Step 1: Using log (CNIC) to define the correlation coefficients (CCs) in countries/regions


The sequential log (CNICs) in each region were extracted from the data [8] to compare the pattern with the CCs between countries/regions (see the MP4 video in Appendix A). For instance, we selected the most similar CNIC patterns (i.e., with higher CCs) for Hubei (China) in Figure 1. It can be seen that Heilongjiang (China), Xinjiang (China), and other regions in China were highlighted in Stage I (e.g., in the first one-third of days during the study period up to 27 March 2020). However, other regions were shown in Stage II (e.g., Guangdong (China) and Guangxi (China)) and Stage III (e.g., Egypt, Iran, Sweden, the Republic of Korea, and Gansu (China)), all of which were dated 27 March 2020. As such, different spread patterns can be examined using the CCs on CNIC across countries/regions.


Step 2: Applying SNA to classify the spread clusters of COVID-19


The CCs for each paired country were included in SNA (see the control file of Pajeck [38] in Appendix A). The higher CCs that a region was associated with, the closer the relationship was within a cluster for regions with identical CNIC patterns.

In keeping with Pajek’s guidelines [38] for SNA, we defined a region as a node (or an actor/vertex) that is connected to another node through an edge (or tie with a line) (i.e., the weight denoted by CC on CNIC between 2 entities). Usually, the weight (dented by CC) is summed (called concentrality degree (CD) by the number of connections between 2 nodes). The more connections (or said co-occurrence) produce a higher CD in the network, the algorithm of the community partitioning method was used to identify and separate clusters in SNA. The number of clusters was set at 3 in the SNA based on Pajeck’s criteria. The SNA file is provided in Appendix A as well.


Step 3: Plotting the SNA to classify the spread clusters of COVID-19


A dashboard was designed for displaying region bubbles colored by cluster types and sized by the weights (i.e., CD denoted by the summation of CCs) on Google Maps. The line (or curve) linked to the 2 nodes stands for the closer relation (or CC > 0.8 or more). The curve disappears if the CC is less than the criterion of 0.8.

Two types of SNA were drawn according to the research design (i.e., one for data from 22 January to 27 March 2020, and another from 1 January to 16 February 2021), including (1) the spread routes in stages, and (2) the SSP of other countries/regions related to Hebei Province (China) and Heilongjiang (China).

### 2.4. Building the Model-Based on IRT

#### 2.4.1. Percentage-Type Observed Data (OPi)

The CNIC in a country/region was transformed into a percentage from 0 to 1 [39,40,41], as shown in Equation (1).
(1)$$\mathrm{OPi}=\frac{\left(\mathrm{Oi}-\mathrm{Min}\right)}{\left(\mathrm{Max}-\mathrm{Min}\right)}\times \mathrm{CRi},$$
where Oi denotes the originally observed case number, and the maximum and minimum are symbolled by Max and Min, respectively. The CR stands for the adjustment coefficient because not all TSCCC equal 1.0 at θ(=5) according to Equation (2) (e.g., at early epidemic stage [32]).

#### 2.4.2. The IRT Probability Model

Two parameters, a and b, mentioned in Section 1.2, were estimated using the IRT model in Equation (2), in which a and b (named delta) represent the discrimination (i.e., the slope shape) and the item difficulty (i.e., the location toward the left (easier or at an earlier stage) or the right (harder or at the later stage in epidemic)) with values within 0 and 3, and −5 and 5, respectively. Theta was set at 5 at the modeling process and used to compute the adjustment coefficient of CRi in Equation (1).
(2)$$P\left(\theta \right)=\frac{\exp(\left(a\left(\theta -\delta \right)\right)}{(1+\exp\left(a\left(\theta -\delta \right)\right)}$$

#### 2.4.3. Ogive Curved in a Model

After model parameters were estimated in Equation (2), the Theta denoted by the location on the scale (Axis X) was determined by Equation (3).
(3)$$\mathrm{Theta}=-5+\left(n_{i-1}\right)\times \frac{\left(5-\left(-5\right)\right)}{N},$$
where *N* is the observed days, and n*_i_* represents *i*th day. The probability (or denoted by the expected percentage, Epi) can be yielded by Equation (2). For instance, the Theta on the day *i* in a total *N* days in epidemic can be obtained by Equation (3).

#### 2.4.4. Transforming Epi into the Number of Expected CNIC

Based on Equation (1), we can obtain the expected CNIC via Equation (4):(4)$$\mathrm{Expected}\;\mathrm{CNIC}=\mathrm{EPi}÷CR_{i}\times \left(\mathrm{Max}-\mathrm{Min}\right)+\mathrm{Min}.$$

### 2.5. Model Parameter Estimation

#### 2.5.1. The Attributes of the Ogive Curve

The shorter epidemic drives the ogive curve toward the left. Otherwise, the longer epidemic drives the curve toward the right side. The length of the IP days affects the NISHC in a specific country/region.

#### 2.5.2. Parameter Estimation

We applied the Microsoft Solver add-in tool to estimate parameters (see the MP4 video in Appendix A).To minimize the total residuals, we used the Microsoft function as shown below.
(5)$$\mathrm{SUMXMY}2\;([\mathrm{OPi}-\mathrm{Epi}]\times [\mathrm{OPi}-\mathrm{Epi}])=\sum_{i=1}^{n}{\left(\mathrm{OPi}-\mathrm{Epi}\right)}^{2}.$$
Estimated parametersIn Equation (2), a and b were estimated.Constrained termsWe set a and b in a range between (0, 4) and (−5, 5), respectively. In addition, the correlation coefficient (CC) between OPi and Epi was set beyond 0.9.Perform the Solver add-inThe Microsoft Solver add-in was performed for each country/region to estimate the model parameters (see Appendix A for more details). The ogive curve can be plotted to predict the future CNIC and determine IP days as explained in the next section.

### 2.6. Determining IP Using a Search Scheme

The IP was determined by the computation of the absolute advantage coefficient (AAC), or the dimension coefficient (DC) [32,39,40,41], in Equation (6).
(6)$$\mathrm{AAC}=\frac{\frac{\gamma_{3}}{\gamma_{2}}}{\frac{\gamma_{2}}{\gamma_{1}}},$$
where AAC is determined by the 3 consecutive Epi, and the IP is located at the minimum one across all possible AACs on an ogive curve [32].

### 2.7. Statistical Tools and Data Analysis

A visual representation displaying the comparison of model parameters among continents/regions was plotted on the Kano diagram [42,43,44]. The IRT-modeling process was executed in Microsoft Excel (see Appendix A). The study flowchart is present in Figure 2.

## 3. Results

### 3.1. Spread Clusters in Color on Google Maps

Three clusters colored by routes and stages are presented in Figure 3. We can see that the spread stages were from Asia to Europe and then to the American continent, as shown by the yellow, green, and red bubbles. Nonetheless, a few bubbles in yellow are located in Canada (North America) and Finland (Europe). Many red bubbles are present across Europe and South America. It is worth noting that Indonesia is red with curved lines (i.e., connected to other associated regions in red) at the middle bottom of Figure 3, which tells us that Indonesia had the same CNIC pattern as the US. Similarly, many red bubbles (e.g., the UK, Ireland, Norway, and Brazil) also had the same CNIC pattern as the US. Readers are invited to scan the QR code in Figure 3 and click on the bubble of interest to inspect the details about regions with similar COVID-19 CNIC patterns, such as those shown in Figure 1.

### 3.2. Details about Indonesia in the CNIC Pattern of COVID-19

We inspected the CNIC patterns in Indonesia from three perspectives at days 49–51, 51–53, and 54–56 (i.e., three stages) in Figure 4. We can see that many CNIC patterns in the US coincided with those in Indonesia, using the closer CC for identification.

### 3.3. Finland’s CNIC Pattern of COVID-19

The CNIC pattern for Finland is presented in Figure 5. We chose Finland because we were curious why just one yellow bubble appeared in Europe. We can see that the CNIC pattern for Finland was much closer to those in yellow areas (e.g., regions in China) and those in green bubbles (e.g., regions in Austria and South America). It is worth noting that a few red bubbles are connected to Finland.

### 3.4. Using an IRT-Based Model to Examine Spread Patterns

Two parameters (i.e., location b and slope a) in an IRT-based COVID-19 model are present in Figure 6, in which bubbles are colored by continents and sized by the number of IP days. Three parts are separated on the Kano diagram, including a higher slope at the top, which is located at the far-right side and neutral in the middle. We can see that COVID-19 spreads from Asia (in yellow on the left) to Oceania (in brown) and Europe (in red), then to North America (in light green), Africa (in darker green), and South America (in black). Bubbles with similar patterns in the CNIC are closer together in Figure 6. For example, in Figure 7, Indonesia is similar to the US states of Wisconsin and Missouri. The observed CNICs are in black, while the predicted CNCIs are in green on the ogive curves. Readers are invited to click on the link [45]. After the bubble of interest is selected, the IRT-based ogive curve, like that in Figure 7, immediately appears on the website.

The comparison of location parameters in continents/countries as of 19 October 2020 is displayed in Table 1. We can see that the means of location parameters were lower China and higher in the US and South America. A significant difference in mean locations on Axis X in Figure 6 was found between China and the US.

### 3.5. The Spread Patterns of COVID-19 in January 2021

To identify countries/regions with the same spread pattern as Hebei Province in China, Figure 8 shows the three spread clusters in color (i.e., all DCCs equal zero in a yellow, mild epidemic in green, and second wave in red). The curved lines shown in Figure 8 are subject to the CC of CNIC between entities greater than 0.98.

Two findings related to the above phenomena were found: (1) Both the provinces of Hebei and Heilongjiang in China had a similar spread pattern in red bubbles; and (2) other countries (or regions), such as Japan, Taiwan, and Qatar, had the same spread pattern as Hebei Province (China) as well. Readers are invited to scan the QR code in Figure 8 and see the blue curved line linked to the bubble of Hebei (China). The country of Qatar is uniquely connected to Hebei (China). The comparison of IP days on the ogive curves is made for each country/region if the bubble is clicked on, such as the one for Hebei (China) shown at the bottom panel in Figure 8. The IRT model in MS Excel is shown in Appendix A.

### 3.6. Online Dashboards Shown on Google Maps

All those plots in Figures appear once the bubble or region of interest is clicked using the links, similar to those deposited in Appendix A.

## 4. Discussion

The classification has been used by humans for thousands of years. It is also important to our everyday life and applies to almost everything we do, allowing us to find and recognize things more easily. Imagine if we went to a library without classification—where would we start looking for a particular book? As of 18 January 2021, over 73,021 articles were found by searching the keyword “classification [title]” in Pubmed. Then, 57 articles were searched by “International journal of environmental research and public health [journal],” such as “machine learning-based activity pattern classification” [46], “support vector machine classification of drunk driving behavior” [47], and “local climate zone classification” [48], to name a few.

### 4.1. Findings and Implications

This study was classified by two periods during the COVID-19 pandemic: (1) From 22 January to 27 March 2020, and (2) from 1 January to 16 February 2021. We observed that (1) the use of SNA for investigating the SSP is viable; (2) the spread routes of COVID-19 from China to West Asia, Europe, North America, and South America are obvious in three classifications (255, n = 51, 130, and 74 in stages); (3) those countries/regions of Heilongjiang (China), Japan, Taiwan, and Qatar are identical in a common cluster with Hebei Province (China), and (4) a dashboard on Google Maps can be applied to display the SSP of the COVID-19 around the world.

### 4.2. What This Finding Adds to What We Already Knew

Many online real- or near-real-time dashboards have been launched for tracking the worldwide spread of the COVID-19 outbreak [2,3,4,5,6,7]. Most of them are similar to other traditional websites [8,9,10,11,12], merely providing the same information as the WHO COVID-Dashboard [5]. The study of COVID-19 requires further mathematical analyses of data regarding cases and data (e.g., deaths and fatality rates) reported worldwide [49]. Although dashboards (e.g., JHU [2] and WHO [5], and others [3,4,6,7]) have provided interesting visualizations for reporting the current state of the COVID-19. However, these presentations lack important information and analysis approaches using mathematical models to predict the projection of DCC or CNIC, which would be useful to understand the trends of COVID-19 [49]. For instance, the COVID-19 Watcher [50] displays detailed COVID-19 data from every county and 188 metropolitan areas in the US. The drawback of it is also similar to other dashboards through different types of charts providing basically necessary information regarding COVID-19.

The COVID-19 Dashboard [49] was designed to offer additional valuable information (e.g., using a mathematical model to project future cases worldwide and by country). The mathematical model is based on linear algebra calculations with a quadratic equation using the last 31-day CNICs. Accordingly, the mathematical model [49] is able to estimate the number of new cases up to later several or more days.

There have been criticisms of the UK government for not providing information on the evidence base used to inform their decision-making [51]. A real-time policy dashboard to aid global transparency in response to COVID-19 is expected to (1) present the localized reasoning behind COVID-19 policy decisions, and (2) allow the global health community to provide further support to governments and international stakeholders [52]. As such, dashboards only presenting real-time descriptions of new daily cases and risk factors are insufficient. A scientific approach involving the mathematical model embedded in the online dashboard is urgently required.

More than 36 articles of COVID-19 using SNA were published up to 16 February 2021 [53]. No authors have applied CCs of CNIC to explore the knowledge of spread patterns in a comparison among countries/regions. The SNA process is simple but useful, and defines the CCs between countries/regions prior to analysis. Two regions with a higher CC have a high probability within an identical cluster. Otherwise, these two will be in two distinctly different clusters [17,18,19,20,21,22,23]. The SSP was thus necessary to use SNA to classify attributes of COVID-19 for countries/regions (e.g., spread stages and occurrences of stationarity in the second wave during the COVID-19 pandemic).

### 4.3. What Is Implied and What Should Be Changed

The visual display, combined with SNA (e.g., in Figure 3 and Figure 5), provides more messages to readers. This is the first study applying SNA to identify SSP of the COVID-19 cases around the world. Two types of SSP were included: (1) Spread stages from 22 January to 27 March 2020 and (2) occurrences of stationarity in the second wave from 1 January to 16 February 2021. These approaches are recommended to future studies and are not just limited to COVID-19 as in this study. The online dashboard involving SNA is as promising and viable as the online SNA illustrated on the website [54].

In addition to many mathematical models [24,25,26,27,28,29,30,31] used for predicting the CNIC in COVID-19, the IRT-based predictive model has been proposed [32]. The IP [13,14,15,16] on the ogive curve should be (and must be) determined before assessing the NISHC. The ogive curve [32] using Equation (5) to determine the IP and IPcase-index [32] evaluated the NISHC, which was different from previous studies [55,56] which have used the DCC or CNIC alone.

The dashboard provided with SNA and IP on the given ogive curve is modern and innovative. Visual displays on CNIC using SSP were developed. We can see that red bubbles are in the UK and the US. The first confirmed case in the UK on 31 January 2020 [57] refers to the SSP in Figure 3 and Figure 5, which is unique when compared to other COVID-19 dashboards [3,4,58].

### 4.4. Strengths of This Study

The current study included three main stages: (1) SNA was used to classify the spread stages and occurrences of stationarity in the second wave during the COVID-19 crisis, (2) comparisons of model parameters (e.g., the location parameter or IP) were made to differentiate epidemic stages, and (3) an app was developed for understanding the transmission patterns of COVID-19 across countries/regions using the mathematical IRT model embedded in dashboards.

### 4.5. Limitations and Future Studies

Our study has some limitations. First, the CC was determined by NCIC between countries/regions. Further studies should compare whether the DCC is replaced by NCIC in the computation of CC.

Second, although we defined the CNIC as the proxy of the confirmed cases on continents to classify the spread patterns using SNA, we suggest that future research should include deaths to compare the difference in SSP.

Third, using the Microsoft Solver add-in to estimate IP days in the IRT model is not a unique approach. Many other methods can be applied to estimation, such as Warm’s weighted mean likelihood estimate [59], anchored maximum likelihood estimation [60], and weighted likelihood estimation [61]. These methods are worthy of comparison in future studies.

Fourth, visual dashboards are shown on Google Maps. However, these installments are not free of charge. For example, the Google Maps application programming interface (API) requires a paid project key for the cloud platform. Thus, the limitations of the dashboard are that it is not publicly accessible, and it is difficult for other authors or programmers to mimic for use in a short period of time.

Fifth, although IRT is common and popular in the educational and psychometric field, many readers in public health are unfamiliar with IRT. The IRT 2-parameter model requires some effort to understand through the data and MP4 videos provided in the Appendix A.

Sixth, only two periods, (1) from 22 January to 27 March 2020 and (2) from 1 January to 16 February 2021, were included in this study. The spread pattern would be varied if different periods were observed because the SARS-CoV-2 virus mutates fast [62,63].

Finally, the mathematical model was proposed in a previous study [49]. Further verification is required to determine whether the model residual is smaller than the IRT model [32]. The comparison study is easy to conduct if the study period is identical (e.g., 31 days [49]) in these two scenarios.

## 5. Conclusions

Two aspects, (1) the spread stages from 22 January to 27 March 2020, and (2) the occurrences of stationarity of second wave in January 2021, were observed using SNA techniques. An app was applied to display the cluster features during the COVID-19 pandemic. More details about the application are given in Appendix A with an MP4 video. An online dashboard developed for displaying the transmission patterns across countries/regions is recommended for future research.

## Figures and Tables

**Figure 1 ijerph-18-02461-f001:**
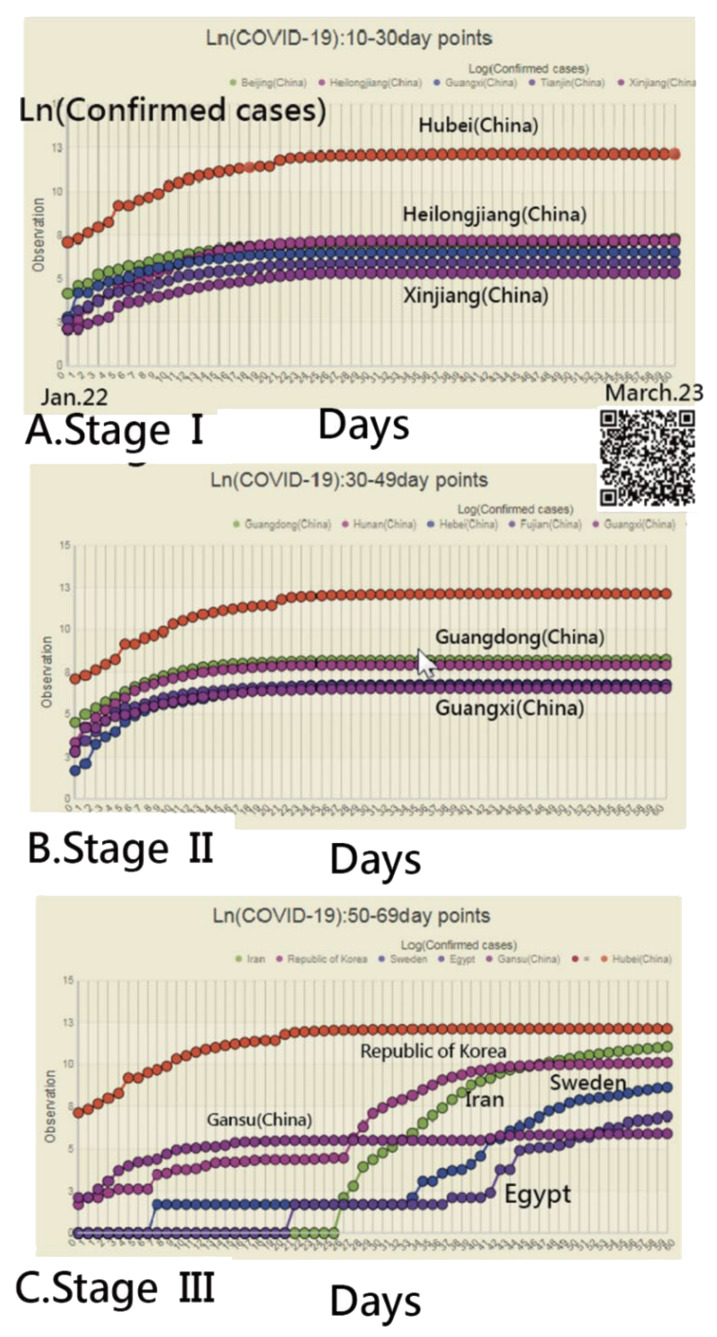
Matching the most similar regions using the correlation pattern across three clusters (stages) (Note: readers are invited to scan the QR-code to examine the detail on a dashboard).

**Figure 2 ijerph-18-02461-f002:**
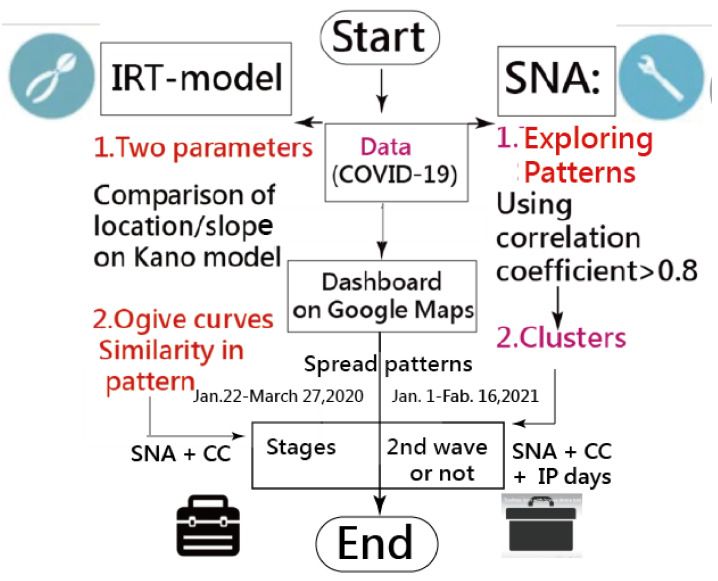
Study flowchart.

**Figure 3 ijerph-18-02461-f003:**
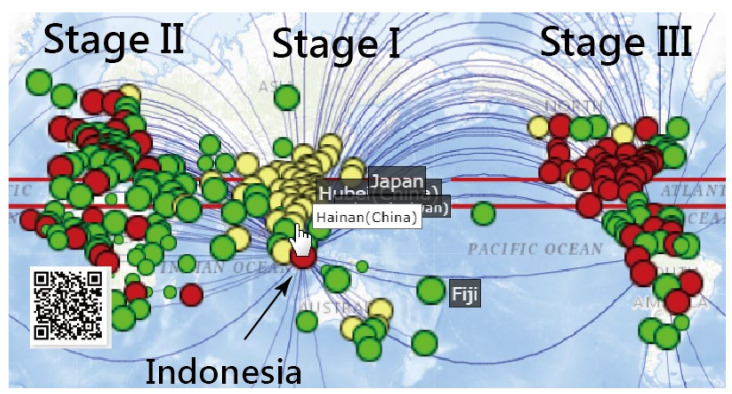
Indonesia compared to three clusters separated by social network analysis (SNA) using the correlation pattern across three stages in yellow, green, and red, respectively. (Note: the curve means the close relation (correlation coefficient (CC) > 0.8) between two countries/regions according to SNA guidelines (Note: readers are invited to scan the QR-code to examine the detail on a dashboard).).

**Figure 4 ijerph-18-02461-f004:**
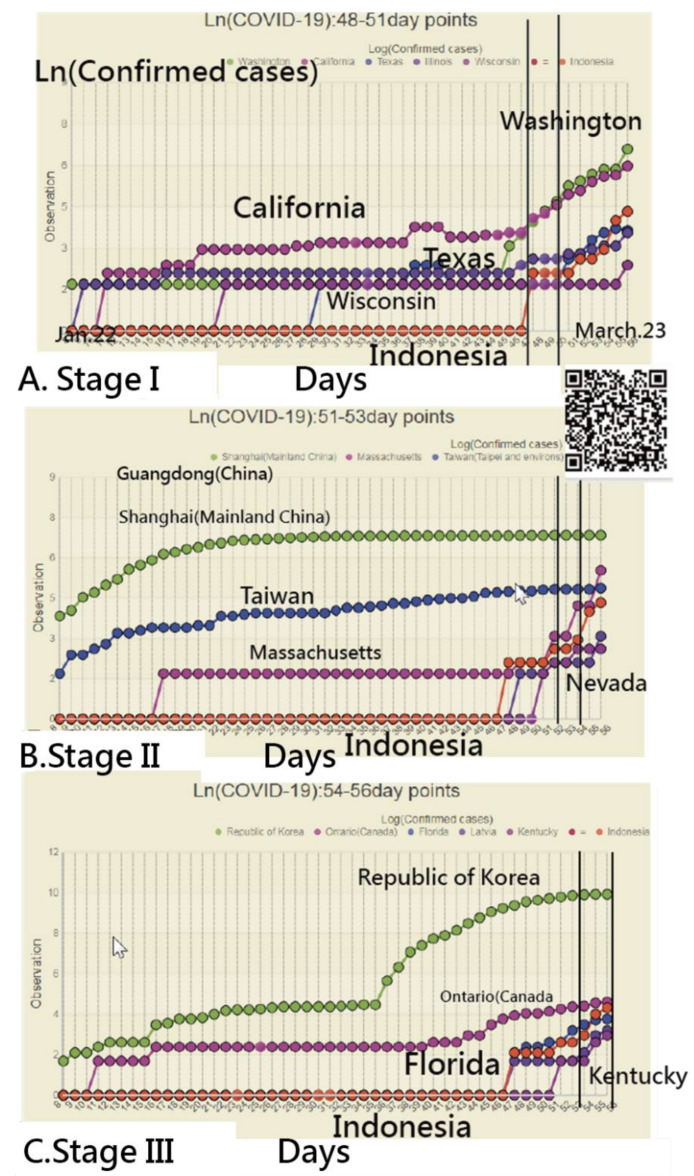
Using the example of Indonesia to interpret the similarities between regions across three stages (Note: readers are invited to scan the QR-code to examine the detail on a dashboard).

**Figure 5 ijerph-18-02461-f005:**
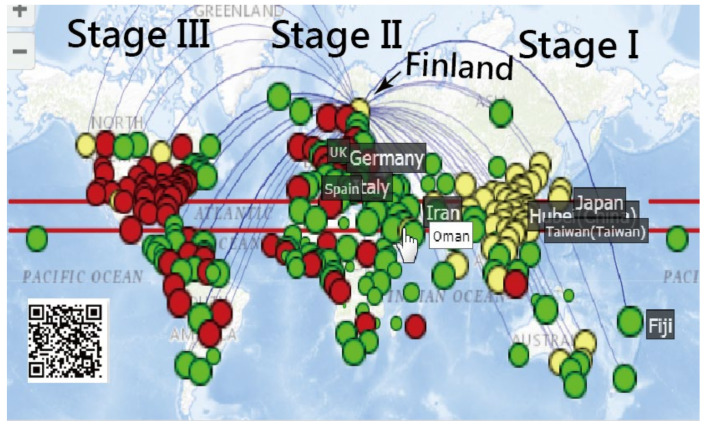
Finland compared to three clusters separated by SNA using the correlation pattern across three stages in yellow, green, and red, respectively. (Note: The curve means the close relation (CC > 0.8) between two countries/regions according to SNA guidelines; readers are invited to scan the QR-code to examine the detail on a dashboard).

**Figure 6 ijerph-18-02461-f006:**
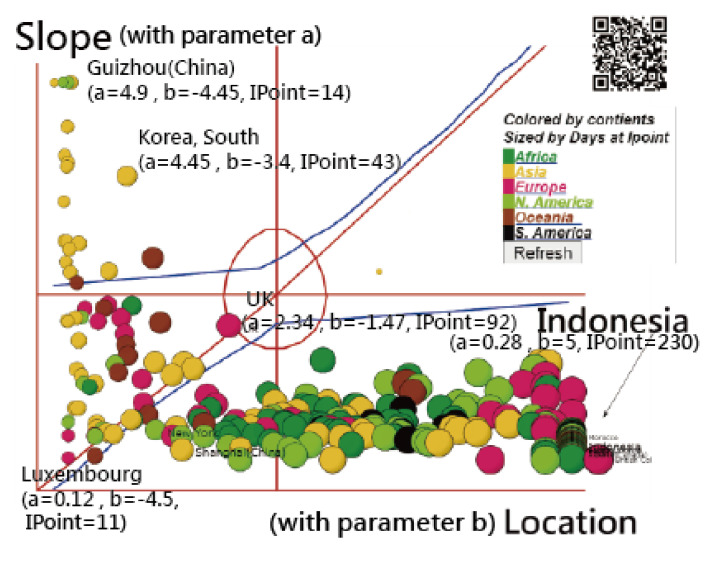
Similarity of the epidemic trend using the item response model (IRT) and Kano models to display the results (Note: readers are invited to scan the QR-code to examine the detail on a dashboard).

**Figure 7 ijerph-18-02461-f007:**
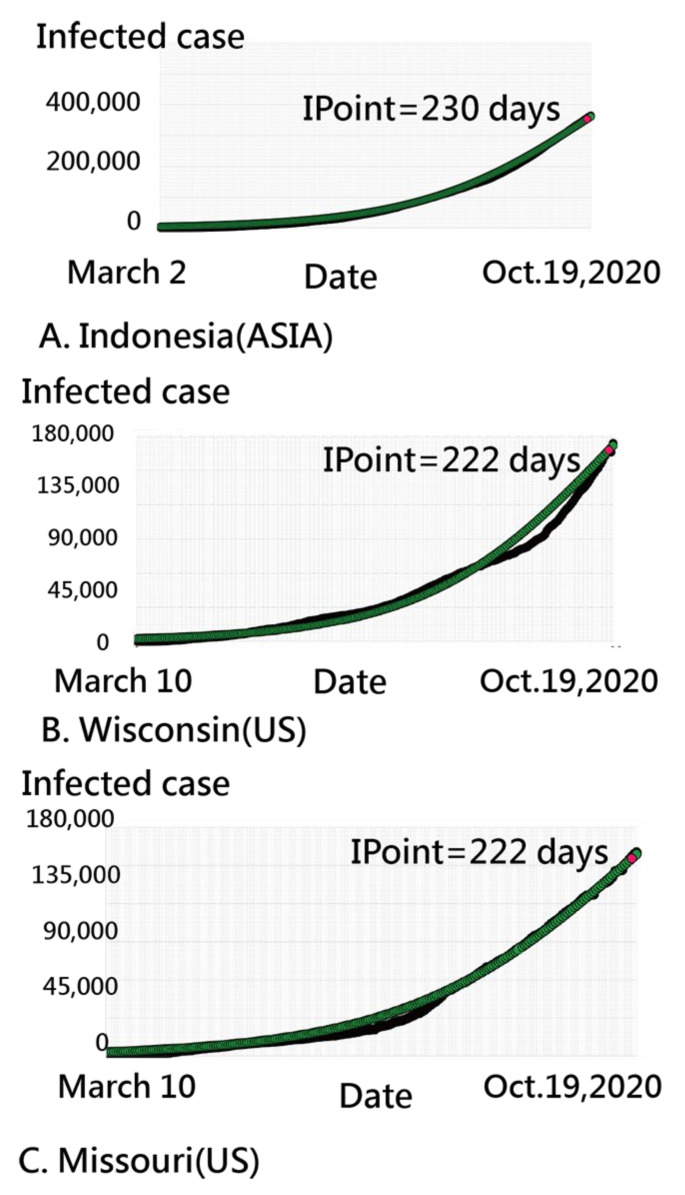
Comparison of epidemic trends displayed as ogive curves.

**Figure 8 ijerph-18-02461-f008:**
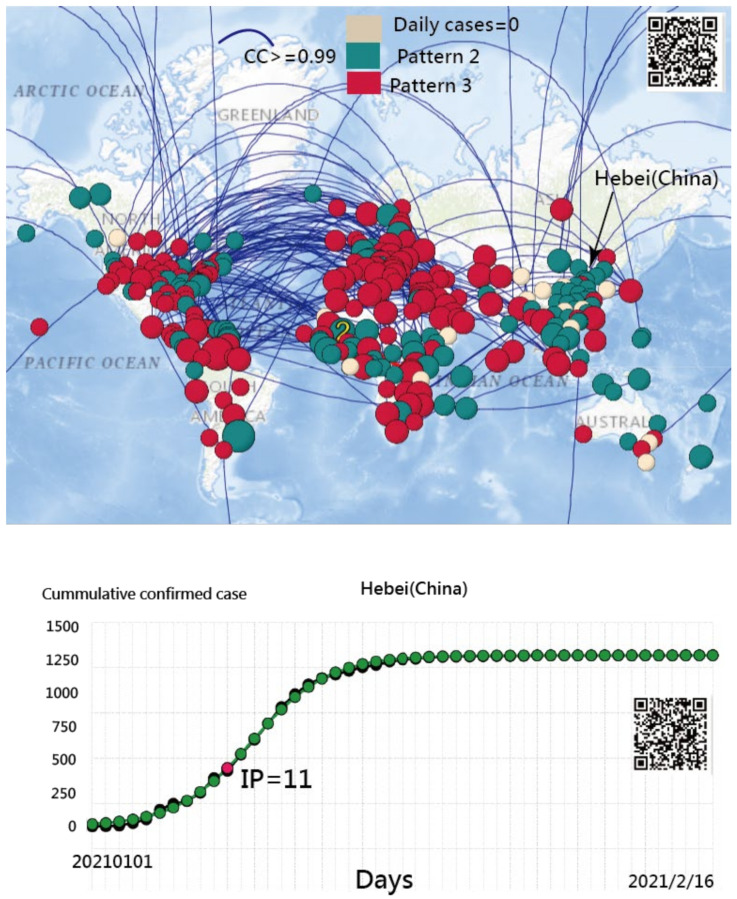
Spread patterns of COVID-19 in countries/regions using CCs (a = 0.99, b = −2.48) (Note: readers are invited to scan the QR-code to examine the detail on a dashboard).

**Table 1 ijerph-18-02461-t001:** Comparison of location parameters in continents and countries as of 19 October 2020.

Area	*n*	Mean	SD	Different (*p* < 0.05)	Stage
				From Area i	
(3) CHINA	31	−2.5382	3.1201	(1)(2)(4)(5)(7)(8)	I
(6) OCEANIA	15	−1.6913	2.4752	(2)(4)(5)(7)(8)	
(1) AFRICA	53	1.0080	2.3828	(3)(4)	II
(2) ASIA	44	1.7731	3.1458	(3)(4)(6)	
(5) N. AMERICA	36	1.9385	3.4053	(3)(4)(6)	
(8) US	61	2.6096	2.5807	(3)(6)	
(7) S. AMERICA	12	2.6469	1.3944	(3)(6)	
(4) EUROPE	49	4.2287	1.9927	(1)(2)(3)(5)(6)	III

Note. F = 22.366, *p* < 0.001.

## Data Availability

All data were deposited in Appendix A.

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
