# Peer review of "Using Social Network Analysis to Identify Spatiotemporal Spread Patterns of COVID-19 around the World: Online Dashboard Development"

_ijerph, 2021, doi:10.3390/ijerph18052461_

Round 1

Reviewer 1 Report

The manuscript by Kyent-Yon Yie et al. is generally well-written and presents a relevant topic of the identification of the spread routes of COVID-19 across continents. The authors mention that the exact routes of the propagation of COVID-19 are still unknown and propose to use a social network analysis (SNA) to determine them. However, they do it for a limited period only (from January 22 to March 27, 2020). The presented results are interesting, but some improvements must be made in the manuscript before it could be accepted for publication. They are specified below:

  • In the abstract you talk about three clusters found in your analysis. However, it’s not clear 3 clusters of what have been found (you need to specify it).
  • On the first page you have a paragraph which is too politicized (“On March 18, 2020, US President Trump defended his increasingly frequent practice of 31 calling the coronavirus the “Chinese Virus”…)

This paragraph should be removed as you are writing a scientific article, which must be free of any political statement.

  • The same concerns a similar paragraph on page 11 “Although Biden’s predecessor Donald Trump repeatedly used terms like “Wuhan virus”…”

This paragraph must be removed as well.

  • In the introduction section you need to cite some useful references discussing geographical and evolutionary origin of the SARS-CoV-2 virus, such as:

For geographical origin:

Kamel Boulos, M.N., Geraghty, E.M. Geographical tracking and mapping of coronavirus disease COVID-19/severe acute respiratory syndrome coronavirus 2 (SARS-CoV-2) epidemic and associated events around the world: how 21st century GIS technologies are supporting the global fight against outbreaks and epidemics. Int J Health Geogr 19, 8

Mercatelli, D., & Giorgi, F. M. (2020). Geographic and genomic distribution of SARS-CoV-2 mutations. Frontiers in microbiology, 11, 1800.

For evolutionary origin:

Makarenkov, V.; Mazoure, B.; Rabusseau, G.; Legendre, P. Horizontal gene transfer and recombination analysis of SARS-CoV-2 genes helps discover its close relatives and shed light on its origin. BMC Ecol. Evol. (2021), 21, 5.

Boni, M.F., Lemey, P., Jiang, X. et al. Evolutionary origins of the SARS-CoV-2 sarbecovirus lineage responsible for the COVID-19 pandemic. Nat Microbiol 5, 1408–1417 (2020).

  • The Conclusion section must be completed by a discussion of possible future work and the comparison of the obtained results with the existing approaches.
  • More details could be given about the application (software) mentioned in the Conclusion section. It should be very useful and helpful for health care portioners.

Author Response

Reviewer 1:

The manuscript by Kyent-Yon Yie et al. is generally well-written and presents a relevant topic of the identification of the spread routes of COVID-19 across continents. The authors mention that the exact routes of the propagation of COVID-19 are still unknown and propose to use a social network analysis (SNA) to determine them. However, they do it for a limited period only (from January 22 to March 27, 2020). The presented results are interesting, but some improvements must be made in the manuscript before it could be accepted for publication. They are specified below:

  • In the abstract you talk about three clusters found in your analysis. However, it’s not clear 3 clusters of what have been found (you need to specify it).
  • Response: In Figure 3, we can see three types of bubbles in color in spread patterns of COVID-19 states from Asia to Europe and America.
  • On the first page you have a paragraph which is too politicized (“On March 18, 2020, US President Trump defended his increasingly frequent practice of 31 calling the coronavirus the “Chinese Virus”…)

This paragraph should be removed as you are writing a scientific article, which must be free of any political statement.

  • Response: As advised by the reviewer, we have removed this paragraph in the revised version of manuscript.
  • The same concerns a similar paragraph on page 11 “Although Biden’s predecessor Donald Trump repeatedly used terms like “Wuhan virus”…”

This paragraph must be removed as well.

  • Response: As advised by the reviewer, we have removed this paragraph in the revised version of manuscript.

In the introduction section you need to cite some useful references discussing geographical and evolutionary origin of the SARS-CoV-2 virus, such as:

For geographical origin:

Kamel Boulos, M.N., Geraghty, E.M. Geographical tracking and mapping of coronavirus disease COVID-19/severe acute respiratory syndrome coronavirus 2 (SARS-CoV-2) epidemic and associated events around the world: how 21st century GIS technologies are supporting the global fight against outbreaks and epidemics. Int J Health Geogr 19, 8

Mercatelli, D., & Giorgi, F. M. (2020). Geographic and genomic distribution of SARS-CoV-2 mutations. Frontiers in microbiology, 11, 1800.

  • Response: We appreciate the suggested references above. The references have been cited in the revised manuscript. In addition, most of parts in Introduction have been rewritten.

For evolutionary origin:

Makarenkov, V.; Mazoure, B.; Rabusseau, G.; Legendre, P. Horizontal gene transfer and recombination analysis of SARS-CoV-2 genes helps discover its close relatives and shed light on its origin. BMC Ecol. Evol. (2021), 21, 5.

Boni, M.F., Lemey, P., Jiang, X. et al. Evolutionary origins of the SARS-CoV-2 sarbecovirus lineage responsible for the COVID-19 pandemic. Nat Microbiol 5, 1408–1417 (2020).

  • Response: The reference has also been cited in the revised manuscript..
  • The Conclusion section must be completed by a discussion of possible future work and the comparison of the obtained results with the existing approaches.
  • Response: The part of conclusion has been rewritten in the revised manuscript to meet the requirement of guideline in scientific article. .
  • More details could be given about the application (software) mentioned in the Conclusion section. It should be very useful and helpful for health care portioners.
  • Response: We have added one sentence in Conclusion: More details about the application are given in Appendix 1 with an MP4 video.

Reviewer 2 Report

Thank you for this manuscript. It is interesting to focus on the spread of Covid-19 via network analysis. The modelling was based on findings from one specific database. It remains unclear whether the data from this database is scientifically sound. As far as can be determined, the data in that particular database originates from countries' statistical offices such as the CDC in the United States of America. This particular agency has provided scientifically highly relevant data since its inception. However, other countries' related agencies may not have such a track record. Hence, the scientific soundness of this database needs to be established. It is equally important to address the fact that the data in the database may not accurately represent the spread of Covid-19 across the world also due to transmission problems, recruitment and recording problems, etc. All these issues need to be carefully addressed. 

Given the fact that the WHO mission to investigate the Covid-19 origin has just ended, it is strongly recommended to rewrite the introductory passage of this manuscript, to avoid all mention of names of specific political personalities. After all, the journal is predominantly scientific and not political. 

Author Response

Reviewer 2:

Thank you for this manuscript. It is interesting to focus on the spread of Covid-19 via network analysis. The modelling was based on findings from one specific database. It remains unclear whether the data from this database is scientifically sound. As far as can be determined, the data in that particular database originates from countries' statistical offices such as the CDC in the United States of America. This particular agency has provided scientifically highly relevant data since its inception. However, other countries' related agencies may not have such a track record. Hence, the scientific soundness of this database needs to be established. It is equally important to address the fact that the data in the database may not accurately represent the spread of Covid-19 across the world also due to transmission problems, recruitment and recording problems, etc. All these issues need to be carefully addressed. 

Response: The data source was from GitHub at https://github.com/CSSEGISandData/2019-nCoV, which is similar to the use of data in Johns Hopkins University Center for Systems Science and Engineering dashboard(JHU). The JHU dashboard’s five authoritative data sources include World Health Organization (WHO), US Centers for Disease Control and Prevention, National Health Commission of the People’s Republic of China, European Centre for Disease Prevention and Control, and the Chinese online medical resource DXY.cn. The dashboard provides links to these sources and others. The corresponding data repository is accessible as Google sheets in GitHub(similar to ours). Please see the statement above in this article at https://www.ncbi.nlm.nih.gov/pmc/articles/PMC7065369/ .

Given the fact that the WHO mission to investigate the Covid-19 origin has just ended, it is strongly recommended to rewrite the introductory passage of this manuscript, to avoid all mention of names of specific political personalities. After all, the journal is predominantly scientific and not political. 

Response: As advised by the reviewer, we have removed this paragraph in the revised version of manuscript.

Reviewer 3 Report

The main goal of the paper is offer an observational analysis for the identification of the spread routes of COVID-19 in the world. 

The data used by the research developed in the paper comes from the freely accessible github records between January 22 and March 27, 2020 largely maintained by a research team from a private company (Google Team).

The analytical methods used are declaratively based on SNA (Social Network Analysis) and IRT (Item Response Theory) however the actual applications are vaguely described and the results summarized in very loose terms.

The presentation of the research is riddled by vague statements, political comments and inconclusive results. In fact there is no rigorous validation of the results but rather a relatively vague description of the rudimentary methods applied.

Overall the manuscript fails to deliver substantially novel and interesting results for the scientific community.

Author Response

Reviewer 3:

 Suggestions for Authors

The main goal of the paper is offer an observational analysis for the identification of the spread routes of COVID-19 in the world. 

The data used by the research developed in the paper comes from the freely accessible github records between January 22 and March 27, 2020 largely maintained by a research team from a private company (Google Team).

The analytical methods used are declaratively based on SNA (Social Network Analysis) and IRT (Item Response Theory) however the actual applications are vaguely described and the results summarized in very loose terms.

Response: We have made a lot of modification in the revised manuscript to make data clear and understandable.

The presentation of the research is riddled by vague statements, political comments and inconclusive results. In fact there is no rigorous validation of the results but rather a relatively vague description of the rudimentary methods applied.

Response: Thanks for giving us another opportunity to revise the manuscript. A lot of changes have been made in the revised version of manuscript. Hopefully, an advanced suggestions would be given to us again as to make the manuscript readable and compresensive to the readership of IJERHR.

Overall the manuscript fails to deliver substantially novel and interesting results for the scientific community.

Response: Many thanks to the reviewer’s comments. The revised manuscript has been improved to illustrate two major parts in COVID-19 periods and present two findings of spread patterns: (1) three stages and (2) three features of stationarity and 2nd wave in COVID-19 using visualizations.

Reviewer 4 Report

Authors have designed a study to evaluate the use of social network analysis on the time series of confirmed cases across continents in order to trace the covid-19 spread routes.

By doing that, authors have demonstrated three different clusters observed by applying social network analysis, of which CCs of Indonesia and United States is found to show similar pattern. Also, a similar pattern is observed for Finland, China, and north America.

An app is developed for better understanding of correlation of the covid-19 cases across different countries/regions. Previous literatures are well cited. But, the introduction and result part of the manuscript can be significantly improved.

Overall, I find this manuscript sound and well conceptualized. It would add valuable information for readers and researchers. However, the following concerns should be addressed by the authors.

  1. Lines 5-12, please fix the authors and their affiliation; affiliation number 4 is not assigned to any author.
  2. Line 13-14, please fix this sentence.
  3. Please fix the spelling of “Stage” in figure 1 and 4 and change the “date” on Axis X to “days” because it’s the number of days between Jan 22 to March 23 is mentioned there. Please keep the same font size of all countries/regions in figure 4 to make it look less crowded. Also, please fix the figure legend of figure 1-6, check whether the font style is consistent with the journal’s requirement.
  4. I do not know what the “slop” implies throughout the manuscript. If the authors mean slope, then please change it accordingly.
  5. Line 111, what does the CR stand for in equ. (1)?
  6. Please provide more information about figure 3 and 5 in the figure legend, such as what are the three different colored bubbles and curved lines imply.
  7. There are many instances where a full stop was not used making sentences incomprehensible. Please fix those errors. For example, in line 169-170, 227, 246, 279-280.
  8. Please fix the references 1, 7, 20, 23, 25, 30-32 by abbreviating journal names. And I am not sure if a link to quora is good way to describe low covid cases in Vietnam in reference 39. Please find other sources to reference it.

Author Response

Reviewer 4:

Authors have designed a study to evaluate the use of social network analysis on the time series of confirmed cases across continents in order to trace the covid-19 spread routes.

By doing that, authors have demonstrated three different clusters observed by applying social network analysis, of which CCs of Indonesia and United States is found to show similar pattern. Also, a similar pattern is observed for Finland, China, and north America.

An app is developed for better understanding of correlation of the covid-19 cases across different countries/regions. Previous literatures are well cited. But, the introduction and result part of the manuscript can be significantly improved.

Response: The introduction has been rewritten in the revised manuscript.

Overall, I find this manuscript sound and well conceptualized. It would add valuable information for readers and researchers. However, the following concerns should be addressed by the authors.

  1. Lines 5-12, please fix the authors and their affiliation; affiliation number 4 is not assigned to any author.

Response: We have added the author affiliation into the list of authorship.

  1. Line 13-14, please fix this sentence.

Response: We have combined the two sentences into one to be correct in grammar. 

  1. Please fix the spelling of “Stage” in figure 1 and 4 and change the “date” on Axis X to “days” because it’s the number of days between Jan 22 to March 23 is mentioned there. Please keep the same font size of all countries/regions in figure 4 to make it look less crowded. Also, please fix the figure legend of figure 1-6, check whether the font style is consistent with the journal’s requirement.

Response: We have revised them according to the reviewer’s advise.

  1. I do not know what the “slop” implies throughout the manuscript. If the authors mean slope, then please change it accordingly.

Response: It is a typo. We have replaced it with slope in the whole revised manuscript.

  1. Line 111, what does the CR stand for in equ. (1)?

Response: We have restated the CR standing for an adjustment coefficient  based on Eq.(1) because not all regions are full in epidemic(e.g., at the first stage) whcn the best-fit model fits to the region’s outbreak of COVID-19.

Details are referred to new references[XXXX].

  1. Please provide more information about figure 3 and 5 in the figure legend, such as what are the three different colored bubbles and curved lines imply.

Response: The legends beneath the two Figure have been added statement” across three stages in colors yellow, green, and red, respectively(note. the curve means the close ralation(CC>0.8) between two countries/regions according to SNA guideline)

  1. There are many instances where a full stop was not used making sentences incomprehensible. Please fix those errors. For example, in line 169-170, 227, 246, 279-280.

Response: The cloud Grammarly software has been applied to check the whole revised manuscript in style, Punctuation marks, etc..

  1. Please fix the references 1, 7, 20, 23, 25, 30-32 by abbreviating journal names. And I am not sure if a link to quora is good way to describe low covid cases in Vietnam in reference 39. Please find other sources to reference it.

Response: We have revised all references according to the reviewer’s advice.

Round 2

Reviewer 2 Report

Thank you for this careful revision of the manuscript. One question remains: Can further rationales for your efforts be found, apart from the one mentioned on P. 2, lines 50-52?

Author Response

Response: We have added one paragraph beneath the part advised by the reviewer: Furthermore, an MP4 video along with the study modules and data in Appendix 1 was provided to readers who are interested in replicating this study on their own, which is rare seen before in the literature.

Reviewer 3 Report

The manuscript has undergone a significant revision. Now it has a clear and rigorously supported scientific message and, in the current form, presents an interesting story to readers in environmental research, geospatial analysis and public health. 

Author Response

Response: Thanks. The most significant contribution is to provide an MP4 video along with study modules and data for readers who are interested in replicating the similar or relevant study in the future at https://osf.io/nyga9/?view_only=8adda14da4a74f218a8bfc2b1cb3939e
